# Protein-Nucleic Acid Complex Modeling with Frame Averaging Transformer

Tinglin Huang[1]*      Zhenqiao Song[2]      Rex Ying[1]      Wengong Jin[3,4]

[1]Yale University, [2]Carnegie Mellon University,
[3]Northeastern University, Khoury College of Computer Sciences
[4]Broad Institute of MIT and Harvard

## Abstract

Nucleic acid-based drugs like aptamers have recently demonstrated great therapeutic potential. However, experimental platforms for aptamer screening are costly, and the scarcity of labeled data presents a challenge for supervised methods to learn protein-aptamer binding. To this end, we develop an unsupervised learning approach based on the predicted pairwise contact map between a protein and a nucleic acid and demonstrate its effectiveness in protein-aptamer binding prediction. Our model is based on FAFormer[2], a novel equivariant transformer architecture that seamlessly integrates frame averaging (FA) within each transformer block. This integration allows our model to infuse geometric information into node features while preserving the spatial semantics of coordinates, leading to greater expressive power than standard FA models. Our results show that FAFormer outperforms existing equivariant models in contact map prediction across three protein complex datasets, with over 10% relative improvement. Moreover, we curate five real-world protein-aptamer interaction datasets and show that the contact map predicted by FAFormer serves as a strong binding indicator for aptamer screening.

## 1 Introduction

Nucleic acids have recently shown significant potential in drug discovery, as shown by the success of mRNA vaccines [26, 61, 60] and aptamers [15, 32, 14, 41]. Aptamers are single-stranded nucleic acids capable of binding to a wide range of molecules, including previously undruggable targets [20, 11]. Currently, aptamer discovery is driven by high-throughput screening, which is time-consuming and labor-intensive. While machine learning can potentially accelerate this process, the limited availability of labeled data presents a significant challenge in ML-guided aptamer discovery [57, 44, 16]. Given this challenge, our goal is to build an unsupervised protein-nucleic acid interaction predictor for large-scale aptamer screening.

Motivated by previous work on unsupervised protein-protein interaction prediction [34], we focus on predicting the contact map between proteins and nucleic acids at the residue/nucleotide level. The main idea is that a predicted contact map offers insights into the likelihood of a protein forming a complex with an aptamer, thereby encoding the binding affinity between them. Concretely, as shown in Figure 1(a), our model is trained to identify specific contact pairs between residues and nucleotides when forming a complex. The maximum contact probability across all pairs is then interpreted as the binding affinity, which is subsequently used for aptamer screening.

---

*Correspondence to `tinglin.huang@yale.edu`
[2]`https://github.com/Graph-and-Geometric-Learning/Frame-Averaging-Transformer`

38th Conference on Neural Information Processing Systems (NeurIPS 2024).

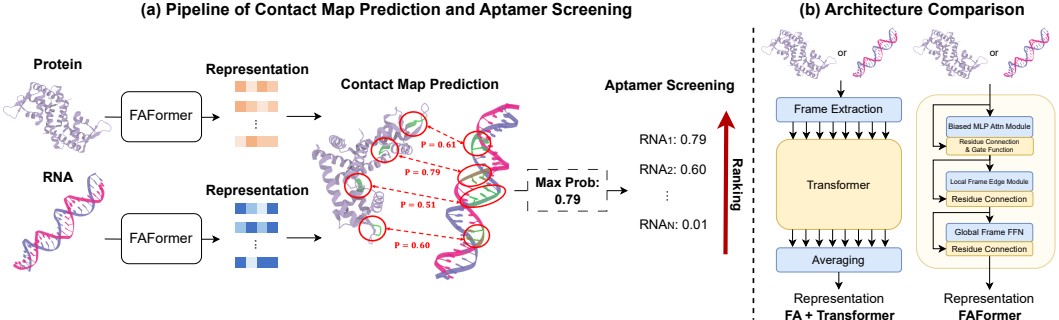

Figure 1: **(a)** The pipeline of contact map prediction between protein and nucleic acid, and applying the predicted results for screening in an unsupervised manner. The affinity score is quantified as the maximum contact probability over all pairs. **(b)** Comparison between Transformer with vanilla frame averaging framework and FAFormer, where the blue cells indicate FA-related modules.

One key factor contributing to the accuracy of contact map prediction models is their capacity to learn equivariant transformations for symmetry groups [67, 39, 66, 74, 52]. A novel line of research focuses on adapting Transformer [83] to equivariant frameworks, leveraging its great expressive power. However, these studies have encountered issues with either (1) high computational overhead with spherical harmonics-based models [47, 27], which complicate encoding by introducing irreducible representations; or (2) limited expressive capability with frame averaging (FA) [63], which diminishes geo-information exploitation by simply concatenating coordinates with node representations.

In light of this, we propose **FAFormer**, an equivariant Transformer architecture that integrates FA as a geometric module within each layer. FA as a geometric component offers flexibility to effectively integrate geometric information into node representations while preserving the spatial semantics of coordinates, eliminating the need for complex geometric feature extraction. FAFormer consists of a *Local Frame Edge Module* that embeds local pairwise interactions between each node and its neighbors; a *Biased MLP Attention Module* that integrates relational bias from edge representation within MLP attention and equivariantly updates coordinates; and a *Global Frame FFN* that integrates geometric features into node representations within the global context.

To validate the advantage of our model architecture, we evaluate FAFormer on two tasks: (1) protein-nucleic acid contact prediction and (2) unsupervised aptamer virtual screening. In the first task, our model consistently surpasses state-of-the-art equivariant models with over 10% relative improvement across three protein complex datasets. For the second task, we collected five real-world protein-aptamer interaction datasets with experimental binding labels. Our results show that FAFormer, trained on the contact map prediction task, is an effective binding indicator for aptamer screening. Compared to RoseTTAFoldNA, a large pretrained model for complex structure prediction, FAFormer achieves comparable performance on contact map prediction and better results on aptamer screening, while offering 20-30x speedup.

## 2   Related Work

**Aptamer screening**   Aptamers are single-stranded RNA or DNA oligonucleotides that can bind various molecules with high affinity and specificity [14, 15, 32, 41]. SELEX (Systematic Evolution of Ligands by EXponential Enrichment) is a conventional technique used for high-throughput screening of aptamers [23, 72, 23], which iteratively selects and amplifies target-bound sequences. Despite its effectiveness, SELEX needs to take substantial time to identify a small number of aptamers [73, 51]. There are some recent studies applying machine learning techniques to aptamer research [18], including generating aptamer structures [36], optimizing SELEX protocols [6], and predicting and modeling protein-aptamer interactions [24, 46, 68, 57]. However, these methods require the user to provide labeled data from time-consuming SELEX assays, thus do not apply to new targets without any SELEX data. Our work focuses on predicting the contact map between protein and nucleic acids using 3D structures and conducting unsupervised screening based on the predicted contact maps, which has not yet been thoroughly explored.

**Protein complex modeling** The prediction and understanding of interactions between proteins and molecules play a crucial role in biomedicine. For example, some prior studies focus on developing a geometric learning method to predict the conformation of a small molecule when it binds to a protein target [71, 19, 53, 50]. As for the protein-protein complex, [25, 29] explore the application of machine learning in predicting the structure of protein multimer. Some studies [77, 56] investigate the protein-protein interface prediction in the physical space. Jin et al. [38] studies the protein-protein affinity prediction in an unsupervised manner. For protein-nucleic acid, some prior works explore the identification of the nucleic-acid-binding residues on protein [65, 86, 89, 37, 85] or predict the binding probability of RNA on proteins [79, 82, 87, 54]. Some previous studies focus on modeling protein-nucleic acid complex by computational method [81, 80, 28]. AlphaFold3 [1] and RoseTTAFoldNA [5] are the recent progresses in this field, which both are pretrained language models for complex structure prediction.

**Geometric deep learning** Recently, geometric deep learning achieved great success in chemistry, biology, and physics domains [13, 90, 40, 55, 52, 64, 10, 70]. The previous methods roughly fall into four categories: 1) Invariant methods extract invariant geometric features from the molecules, such as pairwise distance and torsion angles, to exhibit invariant transformations [67, 31, 30]; 2) Spherical harmonics-based models leverage the functions derived from spherical harmonics and irreducible representations to transform data equivariantly [27, 47, 75]; 3) Some methods encode the coordinates and node features in separate branches, interacting these features through the norm of coordinates [66, 39]; 4) Frame averaging (FA) [63, 21] framework proposes to model the coordinates in eight different frames extracted by PCA, achieving equivariance by averaging the encoded representations.

The proposed FAFormer combines the strengths of FA and the third category of methods by encoding and interacting coordinates with node features using FA-based components. Besides, FAFormer can also be viewed as a combination of GNN and Transformer architectures, as the edge representation calculation functions similarly to message passing in GNN. This integration is made possible by the flexibility provided by FA as an integrated component.

## 3 Frame Averaging Transformer

In this section, we present our proposed FAFormer, a frame averaging (FA)-based transformer architecture. We first introduce the FA framework in Section 3.1 and elaborate on the proposed FAFormer in Section 3.2. Discussion on the equivariance is provided in Section 3.3, and the computational complexity analysis can be found in Appendix B.

### 3.1 Background: Frame Averaging

Frame averaging (FA) [63] is an encoder-agnostic framework that can make a given encoder equivariant to the Euclidean symmetry group. FA applies the principle components derived via Principal Component Analysis (PCA) to construct the frame capable of achieving $E(3)$ equivariance. Specifically, the frame function $\mathcal{F}(\cdot)$ maps a given set of coordinates $\boldsymbol{X}$ to eight transformations:

$$\mathcal{F}(\boldsymbol{X}) = \{(\boldsymbol{U}, \boldsymbol{c}) | \boldsymbol{U} = [\alpha_1 \boldsymbol{u}_1, \alpha_2 \boldsymbol{u}_2, \alpha_3 \boldsymbol{u}_3], \alpha_i \in \{-1, 1\}\} \tag{1}$$

where $\boldsymbol{u}_1, \boldsymbol{u}_2, \boldsymbol{u}_3$ are the three principal components of $\boldsymbol{X}$, $\boldsymbol{U} \in \mathbb{R}^{3\times3}$ denotes the rotation matrix based on the principal components, and $\boldsymbol{c} \in \mathbb{R}^3$ is the centroid of $\boldsymbol{X}$. The main idea of FA is to encode the coordinates as projected by the transformations, followed by averaging these representations. We introduce $f_\mathcal{F}(\cdot)$ to represent the projections of given coordinates via $\mathcal{F}(\cdot)$:

$$\begin{aligned} f_\mathcal{F}(\boldsymbol{X}) &:= \{(\boldsymbol{X} - \boldsymbol{c})\boldsymbol{U} \mid (\boldsymbol{U}, \boldsymbol{c}) \in \mathcal{F}(\boldsymbol{X})\} \\ &:= \{\boldsymbol{X}^{(g)}\}_\mathcal{F} \end{aligned} \tag{2}$$

where $\boldsymbol{X}^{(g)}$ denotes the coordinates transformed by $g$-th transformations. We can apply any encoder $\Phi(\cdot)$ to the projected coordinates and achieve equivariance by averaging, which can be formulated as an inverse mapping $f_{\mathcal{F}^{-1}}(\cdot)$:

$$f_{\mathcal{F}^{-1}}\left(\{\Phi(\boldsymbol{X}^{(g)})\}_\mathcal{F}\right) := \frac{1}{|\mathcal{F}(\boldsymbol{X})|} \sum_g \Phi(\boldsymbol{X}^{(g)})\boldsymbol{U}_g^{-1} + \boldsymbol{c} \tag{3}$$

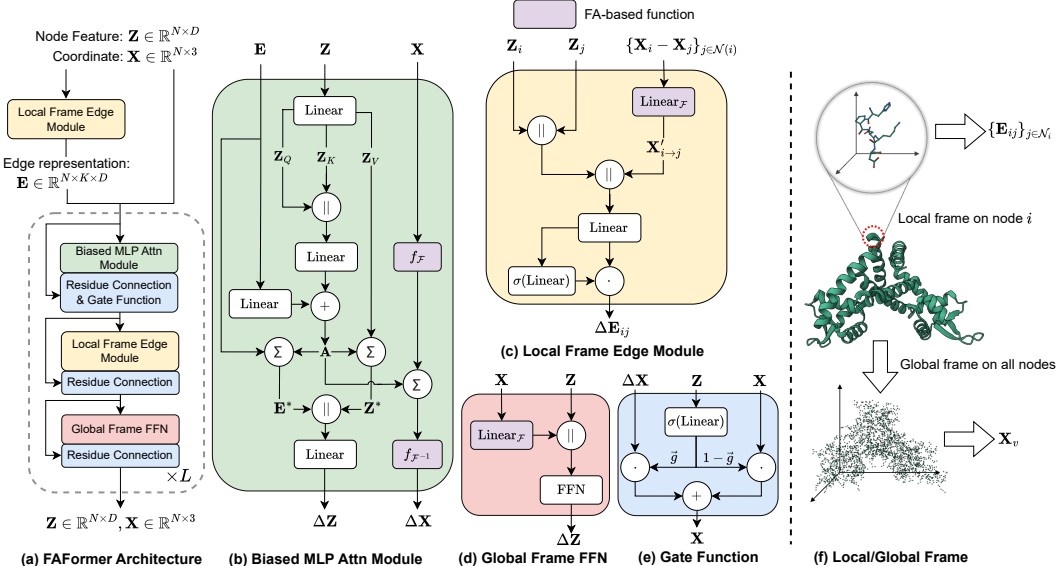

Figure 2: Overview of FAFormer architecture. The input consists of the node features, coordinates, and edge representations, which are processed by a stack of **(b)** Biased MLP Attention Module, **(c)** Local Frame Edge Module, **(d)** Global Frame FFN, and **(e)** Gate Function. $\sum$ deontes aggregation, $\cdot$ is multiplication, $+$ is addition, and $||$ indicates concatenation. Purple cells indicate the operation related to FA. **(f)** illustrates the difference between the local and global frames, where the local frame captures local interactions among the immediate neighbors for each node, while the global frame captures long-range correlations among all nodes.

where $\boldsymbol{U}_g^{-1}$ is the inverse matrix of $g$-th transformations and the result exhibit $E(3)$ equivariance. The outcome is invariant when simply averaging the representations without inverse matrix.

## 3.2 Model Architecture

Instead of serving FA as an external encoder wrapper, we propose instantiating FA as an integral geometric component within Transformer. This integration preserves equivariance and enables the model to encode coordinates effectively in the latent space, ensuring compatibility with Transformer architecture. The overall architecture is illustrated in Figure 2.

**Graph Construction** Each molecule can be naturally represented as a graph [47, 35, 43] where the residues/nucleic acids are the nodes and the interactions between them represent the edges. To efficiently model the macro-molecules (i.e., protein and nucleic acid), we restrict the attention for each node $i$ to its $K$-nearest neighbors $\mathcal{N}_{\text{top-}K}(i)$, within a predetermined distance cutoff $c$:

$$\mathcal{N}(i) = \{j | d_{ij} \leq c \text{ and } j \in \mathcal{N}_{\text{top-}K}(i)\} \tag{4}$$

where we use $\mathcal{N}(i)$ to denote the valid neighbors of node $i$, and $d_{ij}$ denotes the distance between node $i$ and $j$. The long-range context information can be captured by iterative attention within the local neighbors for each node $i$.

**Overall Architecture** As shown in Figure 2(a), the input of FAFormer comprises the node features $\boldsymbol{Z} \in \mathbb{R}^{N \times D}$, coordinates $\boldsymbol{X} \in \mathbb{R}^{N \times 3}$, and edge representations $\boldsymbol{E} \in \mathbb{R}^{N \times K \times D}$ derived by our proposed edge module, where $N$ is the number nodes and $D$ is the hidden size. FAFormer processes and updates the input features at each layer:

$$\boldsymbol{Z}^{(l+1)}, \boldsymbol{X}^{(l+1)}, \boldsymbol{E}^{(l+1)} = f^{(l)}(\boldsymbol{Z}^{(l)}, \boldsymbol{X}^{(l)}, \boldsymbol{E}^{(l)}) \tag{5}$$

where $f^{(l)}(\cdot)$ represents $l$-th layer of FAFormer. In each model layer, we first update coordinates and node features using the Biased MLP Attention Module. Then, these features are fed into Local Frame Edge Module to refine the edge representations. Finally, the node representations undergo further updates through Global Frame FFN.

**FA Linear Module** Based on FA, we generalize the vanilla linear module to encode coordinates in the latent space invariantly:

$$\text{Linear}_{\mathcal{F}}(\boldsymbol{X}) := \frac{1}{|\mathcal{F}(\boldsymbol{X})|} \sum_g \text{Norm}(\boldsymbol{X}^{(g)}) \boldsymbol{W}_g \tag{6}$$

where $\{\boldsymbol{X}^{(g)}\}_{\mathcal{F}}$ is obtained using Equ.(2), $\boldsymbol{W}_g \in \mathbb{R}^{3 \times D}$ is a learnable matrix for $g$-th transformations, and $\text{Norm}(\boldsymbol{X}) := \boldsymbol{X}/\sqrt{\frac{1}{\nu}\|\boldsymbol{X}\|_2^2}$ is the normalization which scales the coordinates such that their root-mean-square norm is one [39]. $\text{Linear}_{\mathcal{F}}(\cdot)$ is an invariant transformation and will serve as a building block within each layer of the model.

**Local Frame Edge Module** We explicitly embed the interactions between each node and its neighbors as the edge representation $\boldsymbol{E} \in \mathbb{R}^{N \times K \times D}$, where $K$ is the number of the neighbors. It encodes the relational information and represents the bond interaction between nodes, which is critical in understanding the conformation of molecules [40, 4, 35].

As shown in Figure 2(f), unlike the vanilla FA which *globally* encodes the geometric context of the entire molecule, the edge module builds frame *locally* around each node's neighbors. Specifically, given a node $i$ and its neighbor $j \in \mathcal{N}(i)$, the geometric context is encoded within the local neighborhood:

$$\{\boldsymbol{X}'_{i \to j}\}_{\mathcal{N}(i)} = \text{Linear}_{\mathcal{F}}\left(\{\boldsymbol{X}_i - \boldsymbol{X}_j\}_{\mathcal{N}(i)}\right) \tag{7}$$

where $\{\boldsymbol{X}_i - \boldsymbol{X}_j\}_{\mathcal{N}(i)}$ denotes the direction vectors from center node $i$ to its neighbors, and $\boldsymbol{X}'_{i \to j} \in \mathbb{R}^d$ is the encoded representation. With the local frame, the spatial information sent from one source node depends on the target node, which is compatible with the attention mechanism. Then the node features are engaged with geometric features, and the edge representation is finalized through the residual connection with the gate mechanism:

$$\boldsymbol{m}_{ij} = \text{Linear}(\boldsymbol{Z}_i || \boldsymbol{Z}_j || \boldsymbol{X}'_{i \to j}) \quad \text{and} \quad \boldsymbol{E}'_{ij} = \boldsymbol{g}_{ij} \cdot \boldsymbol{m}_{ij} + \boldsymbol{E}_{ij} \tag{8}$$

where $(\cdot || \cdot)$ is the concatenation operation, and the calculated gate $\boldsymbol{g}_{ij} = \sigma(\text{Linear}(\boldsymbol{m}_{ij}))$ provides flexibility in regulating the impact of updated edge representation.

The encoded edge representation in FAFormer plays a crucial role in modeling the pairwise relationships between nodes, especially for nucleic acid due to their specific base pairing rules [58, 45]. The incorporation of FA facilitates the encoding of the pairwise relationships in a geometric context, resulting in an expressive representation.

**Biased MLP Attention Module** As shown in Figure 2(b), the attention module of FAFormer first transforms the node features $\boldsymbol{Z}$ into query, key, and value representations:

$$\boldsymbol{Z}_Q = \boldsymbol{Z}\boldsymbol{W}_Q, \ \boldsymbol{Z}_K = \boldsymbol{Z}\boldsymbol{W}_K, \ \boldsymbol{Z}_V = \boldsymbol{Z}\boldsymbol{W}_V \tag{9}$$

where $\boldsymbol{W}_Q, \boldsymbol{W}_K, \boldsymbol{W}_V \in \mathbb{R}^{D \times D}$ are the learnable projections. We adopt MLP attention [12] to derive the attention weight between node pairs, which can effectively capture any attention pattern. The relational information from the edge representation is integrated as an additional bias term:

$$a_{ij} = \text{Softmax}_i\left(\text{Linear}(\boldsymbol{Z}_{Q,i} || \boldsymbol{Z}_{K,j}) + b_{ij}\right), \tag{10}$$

where $b_{ij} = \text{Linear}(\text{LN}(\boldsymbol{E}_{ij}))$ represents the scalar bias term based on the edge representation, $a_{ij}$ denotes the attention score between $i$-th and $j$-th nodes, $\boldsymbol{Z}_{*,i}$ is $i$-th representation of the matrix $\boldsymbol{Z}_*$, $\text{Softmax}_i(\cdot)$ is the softmax function operated on the attention scores of node $i$'s neighbors, and $\text{LN}(\cdot)$ is layernorm function [3].

Besides the value embeddings, the edge representation will also be aggregated to serve as the context for the update of node feature in FAFormer:

$$\boldsymbol{Z}_i^* = \sum_{j \in \mathcal{N}(i)} a_{ij}\boldsymbol{Z}_j, \boldsymbol{E}_i^* = \sum_{j \in \mathcal{N}(i)} a_{ij}\boldsymbol{E}_{ij}, \tag{11}$$

$$\boldsymbol{Z}_i' = \text{LN}\left(\text{Linear}(\boldsymbol{Z}_i^* || \boldsymbol{E}_i^*)\right) + \boldsymbol{Z}_i \tag{12}$$

where $\boldsymbol{Z}_i'$ is the update representation of node $i$. The above attention can be extended to a multi-head fashion by performing multiple parallel attention functions. For the coordinates, we employ an equivariant aggregation function that supports multi-head attention:

$$\boldsymbol{X}^* = f_{\mathcal{F}^{-1}}\left(\{[\boldsymbol{A}^{(0)}\boldsymbol{X}^{(g)}, \cdots, \boldsymbol{A}^{(H)}\boldsymbol{X}^{(g)}]\boldsymbol{W}\}_{\mathcal{F}}\right) \tag{13}$$

where $\{\boldsymbol{X}^{(g)}\}_{\mathcal{F}} = f_{\mathcal{F}}(\boldsymbol{X})$, $H$ is the number of attention heads, $[\cdot]$ is the tensor stack operation and $\boldsymbol{W} \in \mathbb{R}^{H \times 1}$ is a linear transformation for aggregating coordinates in different heads. An additional gate function that uses node representations as input to modulate the aggregation is applied:

$$\boldsymbol{X}' = \boldsymbol{g}_{\text{attn}} \cdot \boldsymbol{X}^* + (1 - \boldsymbol{g}_{\text{attn}}) \cdot \boldsymbol{X} \tag{14}$$

where $\boldsymbol{g}_{\text{attn}} = \sigma(\text{Linear}(\boldsymbol{Z}))$ is the vector-wise gate designed to modulate the integration between the aggregated and the original coordinates. This introduced gate mechanism further encourages the communication between node features and geometric features.

**Global Frame FFN**  To further exploit the interaction between node features and coordinates, we extend the conventional FFN to *Global Frame* FFN which integrates spatial locations with node features through FA, which is illustrated in Figure 2(d):

$$\boldsymbol{X}_v = \text{Linear}_{\mathcal{F}}(\boldsymbol{X}) \quad \text{and} \quad \boldsymbol{Z}' = \text{FFN}(\boldsymbol{Z}||\boldsymbol{X}_v) + \boldsymbol{Z} \tag{15}$$

where $\text{FFN}(\cdot)$ denotes a two-layer fully connected feed-forward network. This integration of spatial information $\boldsymbol{X}_v$ into the feature vectors enables the self-attention mechanism to operate in a geometric-aware manner. Unlike the edge module that focuses on each node's local neighbors, global frame FFN encodes the coordinates of all nodes, thereby capturing the long-range correlation among nodes.

## 3.3   Equivariance

The function $\text{Linear}_{\mathcal{F}}(\cdot)$ exhibits invariance since results are simply averaged across different transformations. In light of this, the node representation generated by the edge module and FFN are also invariant. The biased attention is based on the scalar features so the output is always invariant.

The update of coordinates within FAFormer leverages a multi-head attention aggregation with a gate function. Both functions are $E(3)$-equivariant: attention aggregation is based on frame averaging, while gate function is linear and also exhibits $E(3)$-equivariance, with a formal proof in Appendix C.

In conclusion, FAFormer is symmetry-aware which exhibits invariance for node representations and $E(3)$-equivariance for coordinates.

## 4   Experiments

In this section, we present three protein complex datasets and five aptamer datasets to explore protein complex interactions and evaluate the effectiveness of FAFormer. More details regarding the experiments and datasets can be found in Appendix A and D. Additional experiments, including the ablation studies, and the comparison with AlphaFold3, can be found in Appendix E. All the datasets used in this study are included in our anonymous repository.

### 4.1   Dataset

**Protein Complexes**  We cleaned up and constructed three 3D structure datasets of protein complexes from multiple sources [8, 7, 2, 77]. A residue-nucleotide pair is determined to be in contact if any of their atoms are within 6Å from each other [77, 78]. We conduct dataset splitting based on the protein sequence identity, using a threshold of 50% for protein-RNA/DNA complexes[3] and 30% for protein-protein complexes. The details of all datasets are shown in Table 1.

The protein's and nucleic acid's structures in the validation/test sets are generated by ESMFold [48] (proteins) or RoseTTAFoldNA (nucleic acids). This offers a more realistic scenario, given that the crystal structures are often unavailable.

---

[3]Note that we don't use 30% as the threshold since it results in a very limited validation and test set.

Table 1: Protein complex dataset statistics.

|  | #Train | #Val | #Test | Label |
|---|---|---|---|---|
| Protein-RNA | 1,009 | 115 | 118 | 1.517% |
| Protein-DNA | 2,590 | 134 | 134 | 1.215% |
| Protein-Protein | 4,402 | 544 | 545 | 0.469% |

Table 2: Aptamer dataset statistics.

| Target | GFP | NELF | HNRNPC | CHK2 | UBLCP1 |
|---|---|---|---|---|---|
| #Positive | 520 | 797 | 233 | 1,255 | 892 |
| #Candidate | 1,875 | 9,833 | 3,328 | 10,000 | 10,000 |

**Aptamers**  Our aptamer datasets come from the previous studies [76, 46, 73], including five protein targets and their corresponding aptamer candidates. The affinity of each candidate to the target is experimentally determined. Each dataset is equally split into validation and test sets.

We construct the protein-RNA training set by excluding complexes from our collected dataset with over 30% protein sequence identity to these protein targets, resulting in 1,238 training cases. The 3D structures of proteins are obtained from AlphaFold Database [5], and the structures of RNAs are generated by RoseTTAFoldNA without using MSAs. The statistics are presented in Table 2.

**Feature**  The coordinates of the $C_\alpha$ atoms from residue and the $C_3$ atoms from nucleotide are used as coordinate features. For node feature generation, we employ ESM2 [49] for proteins and RNA-FM [17] for RNA. The one-hot embedding is utilized as DNA's node feature.

## 4.2  Contact Map Prediction

As shown in Figure 1(a), this task aims to predict the exact contact pairs between protein $\{S_i\}_N$ and nucleic acid $\{S'_j\}_{N'}$ which conducts binary classification over all pairs:

$$\text{Model}(S_i, S'_j) = \left\{ \begin{array}{ll} 1, & S_i \text{ contacts with } S'_j \\ 0, & \text{Other} \end{array} \right. \tag{16}$$

**Baselines**  We compare FAFormer with four classes of methods: 1) Vanilla Transformer [83] which doesn't utilize 3D structure; 2) Spherical harmonics-based models Equiformer [47] and SE(3)Transformer [27]; 3) GNN-based models EGNN [66] and GVP-GNN [39]; 4) Transformer with FA [63]. The protein and nucleic acid will be separately encoded with two encoders to avoid label leakage. The representations of residues and nucleotides are concatenated from all pairs and fed into a MLP classifier to conduct prediction.

**Results**  To comprehensively evaluate the performance of label-imbalanced datasets, we apply F1 and PRAUC as the evaluation metrics. The comparison results are presented in Table 3 from which FAFormer reaches the best performance over all the baselines with a relative improvement over 10%.

Additionally, some geometric methods fail to outperform the vanilla Transformer in certain cases. We attribute this to overfitting on crystal structures during training, which hinders their ability to generalize well to unbound structures during evaluation. Compared with serving FA as an external equivariant framework on Transformer, the performance gain on FAFormer verifies the effectiveness of embedding FA as a geometric component within Transformer.

Table 3: Comparison results on three datasets of contact map prediction task.

|  | Metric | Transformer | SE(3)Transformer | Equiformer | EGNN | GVP-GNN | FA | FAFormer |
|---|---|---|---|---|---|---|---|---|
| Protein-RNA | F1 | $0.1021_{.007}$ | $0.0816_{.001}$ | $0.0990_{.005}$ | $0.1093_{.004}$ | $0.1091_{.010}$ | $\underline{0.1150_{.003}}$ | $\mathbf{0.1284_{.003}}$ |
|  | PRAUC | $\underline{0.1015_{.002}}$ | $0.0881_{.001}$ | $0.0985_{.004}$ | $0.0964_{.002}$ | $0.1008_{.004}$ | $0.0965_{.005}$ | $\mathbf{0.1113_{.004}}$ |
| Protein-DNA | F1 | $0.0963_{.006}$ | $0.0824_{.015}$ | $0.0925_{.010}$ | $0.1208_{.009}$ | $0.1225_{.006}$ | $\underline{0.1283_{.001}}$ | $\mathbf{0.1457_{.005}}$ |
|  | PRAUC | $0.1111_{.002}$ | $0.1022_{.007}$ | $0.0913_{.002}$ | $0.1139_{.010}$ | $\underline{0.1195_{.006}}$ | $0.1092_{.006}$ | $\mathbf{0.1279_{.006}}$ |
| Protein-Protein | F1 | $0.0756_{.004}$ | $0.1147_{.011}$ | $0.1039_{.002}$ | $\underline{0.1461_{.001}}$ | $0.1302_{.001}$ | $0.1011_{.002}$ | $\mathbf{0.1596_{.002}}$ |
|  | PRAUC | $0.0707_{.002}$ | $0.0906_{.008}$ | $0.0834_{.002}$ | $\underline{0.1245_{.001}}$ | $0.1181_{.002}$ | $0.0830_{.001}$ | $\mathbf{0.1463_{.003}}$ |

## 4.3  Binding Site Prediction

In addition to contact map prediction, we examine the nucleic acid binding site prediction task, which is a node-level task, to comprehensively evaluate our model. This task solely takes a protein $\{S_i\}_N$

as input and aims to identify the nucleic-acid-binding residues on the protein:

$$\text{Model}(S_i) = \begin{cases} 1, & S_i \text{ contacts with nucleic acid} \\ 0, & \text{Other} \end{cases} \tag{17}$$

Predicting the nucleic acid binding site offers promising therapeutic potential for undruggable targets by conventional small molecule drug [33, 9, 88], expanding the range of potential therapeutic targets.

**Baselines**    We compare FAFormer with two state-of-the-art geometric deep learning models: Graph-Bind [86] and GraphSite [89] in this task. The protein structure is embedded with the geometric encoder and the residue's representations are fed into a classifier for prediction.

**Results**    F1 and PRAUC are applied as the evaluation metrics and we report the average results over three different seeds in Table 4. We can observe that FAFormer achieves the best performance over all the baselines, demonstrating the effectiveness of FAFormer on nucleic acid-related tasks and modeling the geometric 3D structure.

Table 4: Comparison results on binding site prediction.

|  | Metric | GraphBind | GraphSite | FAFormer |
|---|---|---|---|---|
| Protein-DNA | F1 | $0.492_{004}$ | $0.416_{007}$ | $\mathbf{0.506}_{005}$ |
|  | PRAUC | $0.520_{003}$ | $0.541_{001}$ | $\mathbf{0.549}_{004}$ |
| Protein-RNA | F1 | $0.449_{004}$ | $0.400_{007}$ | $\mathbf{0.472}_{003}$ |
|  | PRAUC | $0.471_{005}$ | $0.479_{001}$ | $\mathbf{0.507}_{004}$ |

## 4.4   Unsupervised Aptamer Screening

This task aims to screen the positive aptamers from a large number of candidates for a given protein target. We quantify the binding affinities between RNA and the protein target as the highest contact probability among the residue-nucleotide pairs. The main idea is that two molecules with a high probability of contact are very likely to form a complex [34]. The models are first trained on the protein-RNA complexes training set using the contact map prediction, then the aptamer candidates are ranked based on the calculated highest contact probabilities.

**Results**    Top-10 precision, Top-50 precision, and PRAUC are used as the metrics. As shown in Table 5, the geometric encoders outperform sequence-based Transformer in most cases, and FAFormer generally reaches the best performance. This demonstrates the great potential of an accurate interaction predictor in determining unsupervisedly promising aptamers.

Table 5: Comparison results of zero-shot aptamer screening.

|  | Metric | Transformer | SE(3)Transformer | Equiformer | EGNN | GVP-GNN | FA | FAFormer |
|---|---|---|---|---|---|---|---|---|
| GFP | Top10 Prec. | $0.2333_{094}$ | $0.2000_{141}$ | $0.3000_{000}$ | $0.2666_{047}$ | $0.3000_{081}$ | $0.3000_{141}$ | $\mathbf{0.4000}_{078}$ |
|  | Top50 Prec. | $0.2733_{073}$ | $0.2666_{061}$ | $0.3666_{024}$ | $0.3400_{081}$ | $0.3799_{082}$ | $0.3333_{033}$ | $\mathbf{0.4133}_{041}$ |
|  | PRAUC | $0.2881_{018}$ | $0.2883_{015}$ | $0.3106_{005}$ | $0.3076_{013}$ | $0.3170_{071}$ | $0.2895_{014}$ | $\mathbf{0.3224}_{004}$ |
| HNRNPC | Top10 Prec. | $0.0000_{000}$ | $0.1000_{081}$ | $0.2333_{124}$ | $0.2666_{124}$ | $0.2333_{124}$ | $0.0666_{047}$ | $\mathbf{0.3333}_{160}$ |
|  | Top50 Prec. | $0.0266_{018}$ | $0.1533_{052}$ | $0.1666_{065}$ | $0.2266_{047}$ | $0.2266_{047}$ | $0.1533_{024}$ | $\mathbf{0.2399}_{043}$ |
|  | PRAUC | $0.0641_{005}$ | $0.1178_{016}$ | $0.1191_{033}$ | $0.1525_{010}$ | $0.1434_{035}$ | $0.0913_{003}$ | $\mathbf{0.1628}_{080}$ |
| NELF | Top10 Prec. | $0.1666_{124}$ | $0.2000_{134}$ | $0.1666_{124}$ | $0.2000_{141}$ | $0.0666_{041}$ | $0.1666_{124}$ | $\mathbf{0.2333}_{041}$ |
|  | Top50 Prec. | $0.1866_{049}$ | $0.1599_{041}$ | $0.2000_{033}$ | $0.1333_{037}$ | $0.1133_{061}$ | $0.1533_{049}$ | $\mathbf{0.2399}_{032}$ |
|  | PRAUC | $0.0972_{003}$ | $0.0931_{006}$ | $\mathbf{0.1065}_{003}$ | $0.0969_{013}$ | $0.0850_{005}$ | $0.0982_{001}$ | $0.0963_{008}$ |
| CHK2 | Top10 Prec. | $0.1000_{081}$ | $0.1666_{047}$ | $0.2000_{000}$ | $0.1333_{124}$ | $\mathbf{0.2666}_{124}$ | $0.1666_{047}$ | $0.1000_{081}$ |
|  | Top50 Prec. | $0.1066_{037}$ | $0.0933_{009}$ | $0.1533_{047}$ | $0.1199_{032}$ | $0.1466_{033}$ | $0.1333_{049}$ | $\mathbf{0.1733}_{037}$ |
|  | PRAUC | $0.1273_{004}$ | $0.1253_{003}$ | $0.1271_{003}$ | $0.1268_{002}$ | $0.1249_{003}$ | $0.1251_{005}$ | $\mathbf{0.1297}_{005}$ |
| UBLCP1 | Top10 Prec. | $0.1000_{000}$ | $0.0599_{043}$ | $0.0666_{047}$ | $0.1266_{009}$ | $0.0799_{032}$ | $0.1000_{000}$ | $\mathbf{0.1800}_{009}$ |
|  | Top50 Prec. | $0.1400_{014}$ | $0.1050_{036}$ | $0.1266_{024}$ | $0.1149_{007}$ | $0.1116_{010}$ | $0.1133_{047}$ | $\mathbf{0.1500}_{050}$ |
|  | PRAUC | $0.1004_{004}$ | $0.0956_{006}$ | $0.1026_{002}$ | $0.0977_{002}$ | $0.0968_{002}$ | $0.0977_{003}$ | $\mathbf{0.1070}_{001}$ |

## 4.5 Comparison with RoseTTAFoldNA

In this section, we investigate the performance of RoseTTAFoldNA [5] which is a pretrained protein complex structure prediction model and compare it with FAFormer. The performance of FAFormer is evaluated on the individual predicted protein and nucleic acid structures by ESMFold and RoseTTAFoldNA. We additionally test the performance of AlphaFold3 [1] on a subset of the screening tasks due to AlphaFold3 server submission limits (Appendix E).

**Dataset** For the contact map prediction task, we select the test cases used in RoseTTAFoldNA to create the test set, yielding 86 protein-DNA and 16 protein-RNA cases. Furthermore, the complexes from our collected dataset that have more than 30% protein sequence identity to these test examples are removed. This leads to 1,962 training cases for protein-DNA and 1,094 for protein-RNA, which are used for training FAFormer. The MSAs of proteins and RNAs are retrieved for RoseTTAFoldNA.

For the aptamer screening task, we construct a smaller candidate set for each protein target by randomly sampling 10% candidates, given that the inference of RoseTTAFoldNA with MSA searching is time-consuming. The datasets will be equally split into validation and test sets. More details of these datasets can be found in Appendix D.

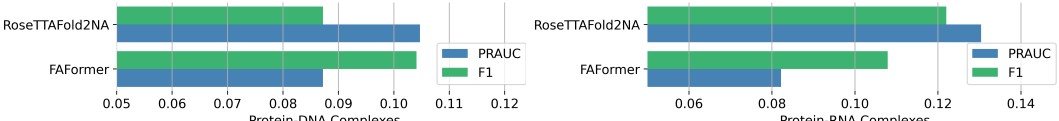

Figure 3: Contact map prediction on RoseTTAFoldNA test set.

Table 6: Comparison results with RoseTTAFoldNA using the sampled datasets, which accounts for the performance differences of FAFormer as shown in Table 5.

|  | Metric | GFP | HNRNPC | NELF | CHK2 | UBLCP1 |
|---|---|---|---|---|---|---|
| RoseTTAFoldNA | Top10 Prec. | 0.4000 | 0.1000 | 0.0 | 0.0 | 0.0 |
|  | Top50 Prec. | 0.3600 | 0.0599 | 0.0 | 0.1000 | 0.0199 |
|  | PRAUC | 0.3926 | 0.1452 | 0.0481 | 0.1176 | 0.0722 |
| FAFormer | Top10 Prec. | $0.4000_{0}$ | $0.1666_{124}$ | $0.1666_{081}$ | $0.1333_{124}$ | $0.1000_{094}$ |
|  | Top50 Prec. | $0.3800_{018}$ | $0.0800_{024}$ | $0.0866_{018}$ | $0.1266_{039}$ | $0.0866_{009}$ |
|  | PRAUC | $0.4027_{022}$ | $0.1781_{089}$ | $0.1044_{018}$ | $0.1374_{013}$ | $0.0762_{016}$ |

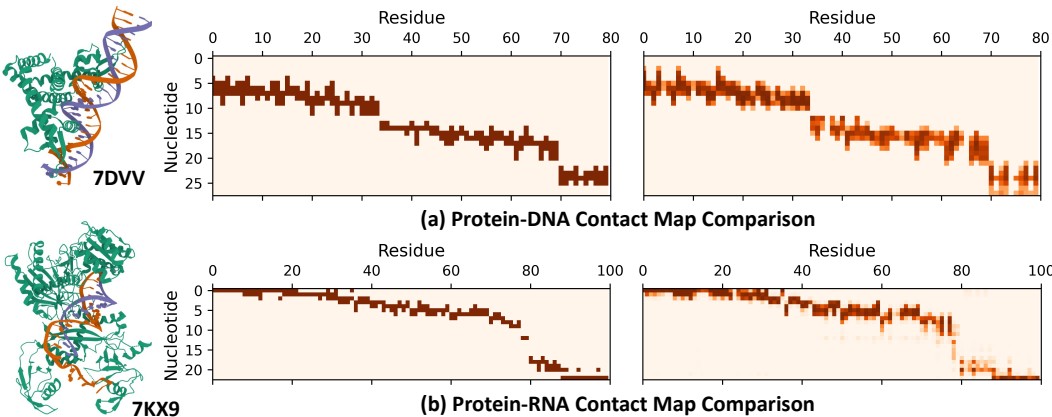

Figure 4: Case study based on two complex examples (PDB id: 7DVV and 7KX9), where each heatmap entry represents the contact probability between the nucleotide and residue. In each row, the figure on the left displays the ground truth contact maps, while the figure on the right displays the results predicted by FAFormer.

**Results** The comparison results of contact map prediction are presented in Figure 3, where FAFormer can achieve comparable performance to RoseTTAFoldNA using unbounded structures.

Specifically, our method has higher F1 scores for protein-DNA complexes (0.103 vs. 0.087) and performs comparably for protein-RNA complexes (0.108 vs. 0.12). Besides, Table 6 shows that RoseTTAFoldNA fails to receive positive aptamers for some targets, e.g., NELF and UBLCP1, while FAFormer consistently outperforms RoseTTAFoldNA for all the targets.

Aligning Figure 3 with Table 6, we observe that while FAFormer does not surpass RoseTTAFoldNA in contact map prediction, it significantly excels in aptamer screening. We attribute this to two reasons: 1) Similar to AlphaFold3 (Table 10), RoseTTAFoldNA as a foundational structure prediction model is optimized for general complex structures, which might bias its performance on some specific protein targets. For example, it can achieve good performance on proteins GFP and HNRNPC but fails for NELF. 2) The protein-RNA test set used by RoseTTAFoldNA for contact map prediction is limited, which may not comprehensively evaluate FAFormer.

**Case Study**    Two examples of protein-DNA (PDB id: 7DVV) and protein-RNA (PDB id: 7KX9) complexes are provided in Figure 4, which shows a visual comparison between the actual (left) and the predicted (right) contact maps. Note that only the residues involved in the actual contact map are presented for a clear demonstration[4]. The complete contact map can be found in Appendix E.1. We can find that despite the sparsity of contact pairs, the predicted contact maps show a high degree of accuracy when compared to the ground truth.

**Time Comparison**    Table 7 shows the average and total inference time of FAFormer and RoseTTAFoldNA on the test cases for the contact map prediction task, including the time for predicting the unbound structures for FAFormer. The predicted structures used for the evaluation of FAFormer are generated without protein and RNA MSAs, demonstrating a significantly faster inference speed by orders of magnitude. For the

|  | Protein-DNA | | Protein-RNA | |
| --- | --- | --- | --- | --- |
|  | Avg. | Total | Avg. | Total |
| RoseTTAFoldNA | 0.175h | 15.12h | 0.440h | 7.04h |
| FAFormer | 32.65s | 0.78h | 51.75s | 0.23h |

Table 7: Inference time on contact map prediction, denoted in seconds ("s") and hours ("h").

unsupervised aptamer screening task, while RoseTTAFoldNA requires only a single MSA search for each protein target, it needs to search MSAs for each RNA candidate sequence separately. Besides, the inclusion of MSAs results in a large input sequence matrix, leading to a time-consuming folding process of RoseTTAFoldNA.

# 5    Conclusion

This research focuses on predicting contact maps for protein complexes in the physical space and reformulates the task of large-scale unsupervised aptamer screening as a contact map prediction task. To this end, we propose FAFormer, a frame averaging-based Transformer, with the main idea of incorporating frame averaging within each layer of Transformer. Our empirical results demonstrate the superior performance of FAFormer on contact map prediction and unsupervised aptamer screening tasks, which outperforms eight baseline methods on all the tasks.

**Broader Impacts**    The proposed paradigm for aptamer screening can be extended to other modalities, such as protein-small molecules and antibody-antigen. Moreover, the strong correlation between contact prediction and affinity estimation demonstrated in our paper can guide future model development. Besides, FAFormer introduces a novel approach to equivariant model design by leveraging the flexibility of FA. This idea opens up numerous possibilities for future research, including exploring different ways to integrate FA with various neural network architectures.

**Limitation**    In this study, the geometric features utilized in the encoders are limited to the coordinates of the $C_\alpha$ and $C_3$ atoms. Features extracted from the backbone or sidechains have not been used, which may limit the performance of the geometric encoders.

---

[4]We sort the residue IDs alongside the row ID so that the contact map appears diagnostic.

# 6 Acknowledgements

We thank Yangtian Zhang, Junchi Yu, Weikang Qiu, and anonymous reviewers for their valuable feedback on the manuscript. This work was supported by the BroadIgnite Award, the Eric and Wendy Schmidt Center at the Broad Institute of MIT and Harvard, NSF IIS Div Of Information & Intelligent Systems 2403317, and Amazon research.

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

# A  Experimental Details

**Running environment.**    The experiments are conducted on a single Linux server with The AMD EPYC 7513-32 Core Processor, 1024G RAM, and 4 Tesla A40-48GB. Our method is implemented on PyTorch 1.13.1 and Python 3.9.6.

**Training details.**    For all the baseline models and FAFormer, we fix the batch size as 8, the number of layers as 3, the dimension of node representation as 64, and the optimizer as Adam [42]. Binary cross-entropy loss is used for contact map identification tasks with a positive weight of 4. The gradient norm is clipped to 1.0 in each training step to ensure learning stability. We report the model's performance on the test set using the best-performing model selected based on its performance on the validation set. All the results are reported based on three different random seeds.

The learning rate is tuned within {1e-3, 5e-4, 1e-4} and is set to 1e-3 by default, as it generally yields the best performance. For each model, we search the hyperparameters in the following ranges: dropout rate in [0, 0.5], the number of nearest neighbors for the GNN-based methods in {10, 20, 30}, and the number of attention heads in {1, 2, 4, 8}. The hyperparameters used in each method are shown below:

- FAFormer: The number of attention heads, dropout rate, and attention dropout rate are 4, 0.2, and 0.2 respectively. We initialize the weight of the gate function with zero weights, and bias with a constant value of 1, ensuring a mostly-opened gate. SiLU [22] is used as the activation function. The distance threshold $c$ is set as 1e5Å and the number of neighbors is 30.

- GVP-GNN[5]: The dimensions of node/edge scalar features and node/edge vector features are set as 64 and 16 respectively. The dropout rate is fixed at 0.2. For a fair comparison, we only extract the geometric feature based on $C_\alpha$, i.e., the forward and reverse unit vectors oriented in the direction of $C_\alpha$ between neighbor residues.

- EGNN[6]: The number of neighbors is set as 30. Besides, we apply gate attention to each edge update module and residue connection to the node update module. SiLU [22] is used as the activation function.

- Equiformer[7] and SE(3)Transformer[8]: The number of attention heads, and the hidden size of each attention head are set as 4 and 16. We exclude the neighbor nodes with a distance greater than 100Å and set the number of neighbors as 30. Based on our experiments, we set the degree of spherical harmonics to 1, as higher degrees tend to lead to performance collapse according to our experiments.

- Transformer and FA[9]: The Transformer is applied as FA's backbone encoder. The dropout and attention dropout rates are 0.2 and 0.2. The number of attention heads is set as 4.

- GraphBind[10]: The dropout ratio and the number of neighbors are 0.5 and 30. We apply addition aggregation to the node and edge update module, following the suggested setting presented in the paper.

- GraphSite[11]: The number of neighbors and dropout ratio are 30 and 0.2. The number of attention layers and attention heads are 2 and 4 respectively. Besides, we additionally use the DSSP features as the node features, as suggested in the paper.

- RoseTTAFoldNA[12]: We employ the released pretrained weight of RoseTTAFoldNA, and set up all the required databases, including UniRef, BFD, structure templates, Rfam, and RNAcentral following the instructions.

---

[5]https://github.com/drorlab/gvp-pytorch
[6]https://github.com/vgsatorras/egnn
[7]https://github.com/atomicarchitects/Equiformer
[8]https://github.com/FabianFuchsML/se3-transformer-public
[9]https://github.com/omri1348/Frame-Averaging
[10]http://www.csbio.sjtu.edu.cn/bioinf/GraphBind/sourcecode.html
[11]https://github.com/biomed-AI/GraphSite
[12]https://github.com/uw-ipd/RoseTTAFold2NA

## B    Efficiency Analysis

### B.1    Computational Complexity

As a core component in the model, frame averaging's time complexity mainly comes from the calculation of PCA among the input coordinates. This operation is practically efficient due to the low dimensionality of the input (only 3 for coordinates). Besides, the calculation of eigenvalue decomposition can be significantly accelerated by some libraries, such as PyTorch [62] and SciPy [84]. We ignore the complexity used for calculating projected coordinates in the following analysis.

As for the local frame edge module in FAFormer, linear transformations are employed to compute the pairwise representation (Equ. (7) and Equ. (8)), and an additional gate is applied to regulate these messages (Equ. (8)), resulting in a computational complexity of $O(NKD + NKD^2)$. Considering the residue connection, the overall complexity of the local frame edge module is $O(NKD + NKD^2 + ND)$.

As for the self-attention module, linear transformations are performed on token embeddings and edge representations (Equ. (9)), and the multi-head MLP attention computation is limited to nearest neighbors (Equ. (10)), leading to the complexity of $O(NHD^2 + NKHD)$ where $H$ is the number of attention heads. Further operations, including aggregation, linear projection, and residual connections (Equ. (11) and Equ. (12)), add $O(NKHD + NKHD^2 + ND)$. Moreover, applying a gate function to the combination of aggregated coordinates and original coordinates (Equ. (13) and Equ. (14)) adds $O(NKH + ND + ND^2)$. As a result, the total complexity for the self-attention module is $O(NKHD + NKHD^2 + NKH + ND + ND^2)$.

Regarding the FFN, most operations are linear transformations that have the complexity of $O(ND^2)$. The gate function and linear combination of coordinates contribute $O(ND^2 + ND)$. So the total computational complexity of FFN is $O(ND^2 + ND)$.

### B.2    Wall-Clock Time Performance Comparison

We conduct a comparison of wall-clock time performance between FAFormer and other geometric baseline models under the same computational environment. Specifically, we measure the average training time for one epoch of each model, with the results illustrated in Figure 5. It can be observed that FAFormer demonstrates greater efficiency compared to spherical harmonics-based models and achieves performance comparable to the GNN-based method GVP-GNN. Such efficiency is attributed to the utilization of top-$K$ neighbor graphs and FA-based modules in FAFormer, which enable efficient modeling of coordinates through linear transformations.

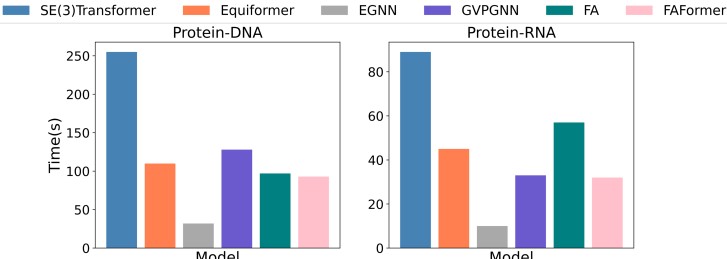

Figure 5: Training time comparison between FAFormer and the other baseline models.

## C    Equivariance Proof

In this section, we investigate the equivariance of the gate function (Figure 2(e)), which can be formulated as:

$$\text{Gate}(\boldsymbol{X}_i, f_\Delta(\boldsymbol{X}_i), \boldsymbol{Z}_i) := \boldsymbol{g}_i \Delta \boldsymbol{X}_i + (1 - \boldsymbol{g}_i)\boldsymbol{X}_i \tag{18}$$

$$:= \boldsymbol{X}_i' \tag{19}$$

where $\boldsymbol{X}_i$ represents the coordinates of $i$-th node, $\boldsymbol{g}_i = \sigma(\text{Linear}(\boldsymbol{Z}_i))$ is the gate score based on $i$-th node's feature, and $\Delta \boldsymbol{X}_i = f_\Delta(\boldsymbol{X}_i)$ denotes the coordinate update function, i.e., the multi-head

aggregation function Equ.(13) which is based on the frame averaging and thus is $E(3)$ equivariant:

$$Q\Delta X_i + t = f_\Delta(QX_i + t) \tag{20}$$

where $t \in \mathbb{R}^3$ is a translation vector and $Q \in \mathbb{R}^{3\times3}$ is an orthogonal matrix.

We aim to prove that the gate function is $E(3)$ equivariant, meaning it is translation equivariant for any translation vector $t \in \mathbb{R}^3$ and rotation/reflection equivariant for any orthogonal matrix $Q \in \mathbb{R}^{3\times3}$. Specifically, we want to show:

$$QX_i' + t = \text{Gate}(QX_i + t, f_\Delta(QX_i + t), Z_i) \tag{21}$$

*Derivation.*

$$
\begin{align}
\text{Gate}(QX_i + t, f_\Delta(QX_i + t), Z_i) &= g_i(Q\Delta X_i + t) + (1 - g_i)(QX_i + t) \tag{22}\\
&= g_i Q\Delta X_i + (1 - g_i)QX_i + g_i t + (1 - g_i)t \tag{23}\\
&= Q\left(g_i \Delta X_i + (1 - g_i)X_i\right) + t \tag{24}\\
&= QX_i' + t \tag{25}
\end{align}
$$

Therefore, we have proven that applying rotation and translation to $X_i$ results in the identical rotation and translation being applied to $X_i'$.

# D  Dataset Descriptions

**Protein Complex Datasets**   We collect the complexes from PDB [8], NDB [7], RNASolo [2] and DIPS [77] databases. Complexes are excluded if they have protein sequences shorter than 5 or longer than 800 residues, or nucleic acid sequences shorter than 5 or longer than 500 nucleotides. The redundant proteins with over 90% sequence similarity to other sequences within the datasets are removed. The protein and the binder structures will be separated and decentered.

**Aptamer Datasets**   Detailed information on each protein target and the aptamer candidates is presented below. The threshold for categorizing sequences as positive or negative aptamers is determined by referencing previous studies or identifying natural cutoffs in the affinity score distributions.

- GFP[13]: Green fluorescent protein. The aptamer candidates are mutants of GFPapt [69], with $K_d$ values ranging from 0nM to 125nM as affinity measures. Candidates with $K_d$ values lower than 10nM are considered positive cases.

- NELF[14]: Negative elongation factor E. The aptamer candidates are mutants of NELFapt [59], with $K_d$ values ranging from 0nM to 183nM as affinity measures. Candidates with $K_d$ values lower than 5nM are considered positive cases.

- HNRNPC[15]: Heterogeneous nuclear ribonucleoproteins C1/C2. The aptamer candidates are the randomly generated RNA 7mers and we apply the affinity values provided by the previous studies [46]. Candidates with affinity scores lower than -0.5 are positive.

- CHK2[16]: Serine/threonine-protein kinase Chk2. Szeto et al [73] applied SELEX (Systematic Evolution of Ligands by EXponential Enrichment) [23] to screen aptamers from a large library of random nucleic acid sequences through multiple rounds. During each round, the bound sequences were amplified and isolated. We use the final round of sequences as the candidates, with sequences having multiplicities over 100 considered positive aptamers.

- UBLCP1[17]: Ubiquitin-like domain-containing CTD phosphatase 1. Similar to CHK2, the final round of sequences are considered candidates, with sequences having multiplicities over 200 considered positive aptamers.

---

[13] https://www.uniprot.org/uniprotkb/P42212/entry
[14] https://www.uniprot.org/uniprotkb/P92204/entry
[15] https://www.uniprot.org/uniprotkb/P07910/entry
[16] https://www.uniprot.org/uniprotkb/O96017/entry
[17] https://www.uniprot.org/uniprotkb/Q8WVY7/entry

Table 8: Sampled aptamer dataset statistics.

| Target | GFP | NELF | HNRNPC | CHK2 | UBLCP1 |
|---|---|---|---|---|---|
| #Positive | 55 | 62 | 20 | 122 | 66 |
| #Candidate | 188 | 981 | 328 | 1,000 | 1,000 |

**Sampled Aptamer Datasets**    We construct smaller aptamer datasets for the comparison between RoseTTAFoldNA and FAFormer by randomly sampling 10% candidates from the original datasets. The statistics are shown in Table 8. We equally split each dataset into a validation set and a test set. The performance of FAFormer on the test set is reported with the best performance on the validation set. RoseTTAFoldNA is directly evaluated on the test set.

**Test datasets of RoseTTAFoldNA**    The test datasets used to evaluate RoseTTAFoldNA are available at this accessible link. We downloaded the dataset and filtered out non-dimer complexes, resulting in 86 protein-DNA and 16 protein-RNA complexes.

# E    Additional Experiments

## E.1    Case Study

Figure 6 presents the complete groundtruth and predicted contact maps of the cases used in Figure 4, where the model accurately captures the sparse pattern.

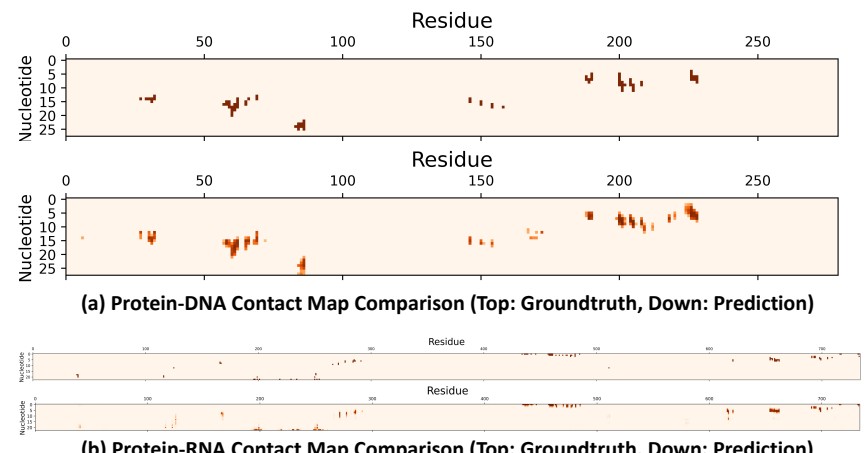

(a) Protein-DNA Contact Map Comparison (Top: Groundtruth, Down: Prediction)

(b) Protein-RNA Contact Map Comparison (Top: Groundtruth, Down: Prediction)

Figure 6: Case study based on two complex examples (PDB id: 7DVV and 7KX9).

## E.2    Ablation Study

In this section, we conduct an ablation study to investigate the impact of FAFormer's core modules. Specifically, we individually disable the edge module and attention mechanism, and replace the proposed FFN with the conventional FFN in FAFormer. The results are presented in Table 9.

Table 9: Ablation study.

| | Protein-DNA | | Protein-RNA | | Protein-Protein | |
|---|---|---|---|---|---|---|
| | F1 | PRAUC | F1 | PRAUC | F1 | PRAUC |
| FAFormer | $0.1457_{005}$ | $0.1279_{006}$ | $0.1284_{003}$ | $0.1113_{004}$ | $0.1596_{002}$ | $0.1463_{003}$ |
| w/o Edge | $0.1171_{004}$ | $0.1225_{002}$ | $0.1048_{007}$ | $0.0983_{008}$ | $0.1100_{005}$ | $0.0972_{002}$ |
| w/o Attention | $0.1401_{001}$ | $0.1250_{006}$ | $0.1059_{002}$ | $0.0973_{001}$ | $0.1325_{001}$ | $0.1121_{001}$ |
| w/o FAFFN | $0.1332_{001}$ | $0.1211_{002}$ | $0.1078_{005}$ | $0.0958_{001}$ | $0.1474_{001}$ | $0.1334_{001}$ |

Removing any core module results in a significant performance decline. Notably, the model degrades to either an attention-only or message-passing-only architecture when the edge or attention module is removed, both of which lead to significant declines in performance. This demonstrates the advantage of combining these two architectures with FA.

### E.3   Aptamer Screening with AlphaFold3

AlphaFold3 [1] represents the latest advancement in biomolecular complex structure prediction and is accessible through an online server[18]. Due to its limited quota (20 jobs per day), we evaluated it on two of the smallest sampled aptamer datasets, GFP and HNRNPC. The statistics for these datasets are presented in Table 8. We used the contact probability produced by the server and the maximum probability as the estimated affinity. The results are shown in Table 10. Although AlphaFold3 performs well in predicting the complex structure, it fails to identify promising aptamers in our cases. We attribute this to fine-grained optimization and overfitting on molecule interaction patterns in the structure prediction task, which have biased its screening performance on specific protein targets.

Table 10: Comparison results with AlphaFold3.

|        | Metric      | AlphaFold3 | RoseTTAFoldNA | FAFormer |
|--------|-------------|------------|---------------|----------|
| GFP    | Top10 Prec. | 0.3000     | 0.4000        | $0.4000_{.0}$ |
|        | Top50 Prec. | 0.3199     | 0.3600        | $0.3800_{.018}$ |
|        | PRAUC       | 0.3132     | 0.3926        | $0.4027_{.022}$ |
| HNRNPC | Top10 Prec. | 0.1000     | 0.1000        | $0.1666_{.124}$ |
|        | Top50 Prec. | 0.0799     | 0.0599        | $0.0800_{.024}$ |
|        | PRAUC       | 0.1355     | 0.1452        | $0.1781_{.089}$ |

---

[18]https://golgi.sandbox.google.com/

