# OpenReview forum: "Protein-Nucleic Acid Complex Modeling with Frame Averaging Transformer"
_NeurIPS.cc/2024/Conference — NeurIPS 2024 poster_

### Official Review · Reviewer_npWJ · 2024-07-09

**Soundness:** 4
**Presentation:** 4
**Contribution:** 3
**Rating:** 7
**Confidence:** 3

**Summary:**

The authors propose am method for predicting contacts between proteins and aptamers based on frame averaging transformers. They showcase their methods on contact prediction and unsupervised aptamer screening, showing improvements over some baselines. The authors also compare to RoseTTAFoldNA, which seems to outperform their method.

**Strengths:**

- Frame averaging is a very interesting idea (published in another venue before) and applying it to this specific problem seems to be very promising
- The presentation is very clear and the paper is well written
- The problem addressed (aptamer screening) is very important and practically relevant
- The authors compare to a large number of different architectures (see Table 4)

**Weaknesses:**

Comparing to a pre-trained structure prediction method is clearly something to be explored well, and the authors do that in section 4.4 (with additional results with AF3 in the appendix). At the same time, this part seems to me not very well developed in the paper. I am especially confused by Figure 3, which the authors comment with "[..] where FAFormer can achieve comparable performance to RoseTTAFoldNA using unbounded structures." Maybe I am not understanding something but the right hand side of the plot seems to clearly indicate that at least for RNAs, RoseTTAFoldNA strongly outperforms the authors' method.

**Questions:**

- Could the authors expand section 4.4 and discuss the combination of Figure 3 and Table 5?
- Do the times in Table 6 include MSA search for Rosetta? If the protein is unchanged,  could you make it more efficient by doing the MSA search only once?
- Could the authors expand a bit on possible future research (e.g. applying the same method to other screening tasks)?

**Limitations:**

Authors discuss some scientific limitations. I do not foresee any negative societal impact.

---

> ### Author Rebuttal · Authors · 2024-08-06
>
> ```
> Weakness: Comparing to a pre-trained structure prediction method is clearly something to be explored well, and the authors do that in section 4.4 (with additional results with AF3 in the appendix). At the same time, this part seems to me not very well developed in the paper. I am especially confused by Figure 3, which the authors comment with "[..] where FAFormer can achieve comparable performance to RoseTTAFoldNA using unbounded structures." Maybe I am not understanding something but the right hand side of the plot seems to clearly indicate that at least for RNAs, RoseTTAFoldNA strongly outperforms the authors' method.
> ```
>
> **Ans**: We apologize for any confusion and will make this clearer in our updated manuscript. Our claim is based on the F1 scores, which show better performance for protein-DNA complexes (0.103 vs 0.087) and comparable performance for protein-RNA complexes (0.108 vs 0.12). It should be noted that the number of test protein-RNA complexes from RoseTTAFoldNA is limited (16 complexes), which might not be able to comprehensively assess our method's performance.
>
> ```
> Question: Could the authors expand section 4.4 and discuss the combination of Figure 3 and Table 5?
> ```
>
> **Ans**: Aligning Figure 3 with Table 5, we observe that while FAFormer does not achieve higher performance than RoseTTAFoldNA in terms of contact map prediction, it significantly outperforms RoseTTAFoldNA in aptamer screening tasks. We attribute this to two main reasons:
>
> - RoseTTAFoldNA as a foundational structure prediction model is optimized for generally predicting the protein-RNA complex structure, which might bias its performance on some specific protein targets. For example, it can achieve good performance on proteins GFP and HNRNPC but fails for NELF.
> - For the contact map prediction comparison, the protein-RNA test set used by RoseTTAFoldNA is limited (16 complexes), which may not comprehensively evaluate the performance of FAFormer.
>
> All these discussions will be added to our revised manuscript.
>
> ```
> Question: Do the times in Table 6 include MSA search for Rosetta? If the protein is unchanged, could you make it more efficient by doing the MSA search only once?
> ```
>
> **Ans**: The times reported in Table 6 include the time required for searching MSAs. For the screening task, RoseTTAFoldNA only needs to search MSAs for protein targets once. However, it still needs to search MSAs for each RNA sequence individually. Besides, the forward process of RoseTTAFoldNA is computationally expensive because the inclusion of MSAs results in a large input sequence matrix. We will add this discussion to our revised manuscript.
>
> ```
> Question: Could the authors expand a bit on possible future research (e.g. applying the same method to other screening tasks)?
> ```
>
> **Ans**: For future applications, the proposed paradigm for aptamer screening can be extended to other modalities, such as protein-small molecules and antibody-antigen. Moreover, the strong correlation between contact prediction and affinity estimation demonstrated in our paper can guide future model development. Specifically, this correlation suggests promising directions for designing new objectives and collecting datasets that better capture the nuances of molecular interactions.
>
> Regarding the architecture, FAFormer introduces a novel approach to equivariant model design by leveraging the flexibility of Frame Averaging (FA). This idea opens up numerous possibilities for future research, including exploring different ways to integrate FA with various neural network architectures.

---

> > ### Comment · Reviewer_npWJ · 2024-08-11
> >
> > I thank the authors for addressing my concerns. My score (accept) remains the same.

---

> > > ### Author Response · Authors · 2024-08-12
> > >
> > > Thank you again for providing valuable comments on our paper! All the suggestions will be added to our updated manuscript.

---

### Official Review · Reviewer_qqru · 2024-07-10

**Soundness:** 2
**Presentation:** 3
**Contribution:** 2
**Rating:** 4
**Confidence:** 3

**Summary:**

This paper mainly focuses on protein-nucleic acid contact prediction and unsupervised aptamer virtual screening. The latter is based on an unsupervised learning approach by predicting the contact maps.  An equivariant architecture that integrates frame averaging and transformer blocks is proposed for this task. Experiments on the two tasks demonstrate the superiority of the proposed architecture compared with other geometric deep learning models.

**Strengths:**

- The proposed architecture is effective on the contact map prediction tasks. The tasks are novel and not well-studied in the community of machine learning.
- The unsupervised learning approach that predicts contact maps is effective on aptamer screening tasks. This perspective is novel.
- The authors have provided codes and reproducibility can be ensured.

**Weaknesses:**

- The novelty of the proposed architecture is limited. The proposed architecture simply combines frame averaging and transformer blocks. The idea that builds the attention module based on SE(3)-invariant features is proposed in Equiformer. The dataset and benchmark track may be more suitable for this paper.
- The expressiveness of the proposed architecture is demonstrated in the contact prediction tasks. More experiments on more general tasks are needed to further show the effectiveness of the proposed architecture, such as protein-protein docking and others.

**Questions:**

See the above weaknesses.

**Limitations:**

The authors have pointed out the limitations due to only modeling certain atoms in the related tasks.

---

> ### Author Rebuttal · Authors · 2024-08-06
>
> ```
> Weakness: The novelty of the proposed architecture is limited. The proposed architecture simply combines frame averaging and transformer blocks. The idea that builds the attention module based on SE(3)-invariant features is proposed in Equiformer. The dataset and benchmark track may be more suitable for this paper.
> ```
>
> **Ans**: We respectfully disagree with the statements and would like to emphasize the contributions of our paper:
>
> First, this is the first work designing a new equivariant Transformer based on the Frame Averaging (FA). Instead of simply combining them, each module is **dedicatedly integrated** with FA, which has **effectively enhanced model performance**, as demonstrated in our ablation study:
>
> - The local frame edge module focuses on local spatial context by constructing the frames on the point cloud centered around each node;
> - The biased MLP attention module applies FA to enable equivariant multi-head attention on the geometric features;
> - Global frame FFN extends the FFN by incorporating geometric information within node representations using FA, allowing the attention to be conducted in a geometry-aware manner.
>
> Second, the way FAFormer achieves equivariance is **completely different** from Equiformer which relies on spherical harmonics operations. As discussed in the Introduction (*Lines 32-38*), models like Equiformer **compromise efficiency** due to the complexity of irreducible representations, as shown in our efficiency comparison (*Appendix B.2*). We believe that incorporating FA within the model opens new possibilities for designing equivariant architectures in this domain.
>
> Third, besides the architectural contributions, we propose **a new paradigm** **for unsupervised aptamer screening**. This paradigm connects contact map prediction and affinity estimation between two molecules, going beyond simply collecting datasets and constructing benchmarks.
>
> ```
> Weakness: The expressiveness of the proposed architecture is demonstrated in the contact prediction tasks. More experiments on more general tasks are needed to further show the effectiveness of the proposed architecture, such as protein-protein docking and others.
> ```
>
> **Ans**: The contact map prediction result between proteins is presented in our paper (*Table 3*). Besides, we have provided results on **binding site prediction** in *Appendix E.1*. This task solely takes a protein as input and aims to predict the NA-binding residues on the protein, which is a node-level prediction task.
>
> To further demonstrate the performance of FAFormer, we here present its performance on two protein understanding tasks, including:
>
> - Fold prediction: This task aims to predict the fold class (a total of 1195 classes) for each protein, with three different test sets (Fold, Family, and Super-Family). Accuracy is used as the evaluation metric.
> - Reaction prediction: This task aims to predict the class of reactions for a given protein catalyzes (a total of 384 classes). Accuracy is used as the evaluation metric.
>
> We follow the experimental settings in ProteinWorkshop[1], including dataset splits and input features (Ca-only). The statistics of the datasets are shown below:
>
> |  | Fold | Reaction |
> | --- | --- | --- |
> | #Train | 12.3K | 29.2K |
> | #Valid | 0.7K | 2.6K |
> | #Test | 1.3/0.7/1.3K | 5.6K |
>
> The comparison results are presented below from which we can observe the best performance of FAFormer across most of the tasks:
>
> ||Fold(Fold)|Fold(Family)|Fold(Superfamily)|Reaction|Average|
> |---|---|---|---|---|---|
> |SchNet|0.2071|0.7670|0.2375|0.5894|0.4502|
> |GearNet|0.3090|0.9340|0.4465|0.7814|0.6177|
> |EGNN|0.2577|0.9193|0.3568|0.6578|0.5479|
> |GCPNet|**0.3276**|0.9359|0.4110|0.6697|0.5860|
> |TFN|0.2512|0.9188|0.3426|0.6922|0.5512|
> |FAFormer|0.2451|**0.9661**|**0.4883**|**0.7970**|**0.6241**|
>
> Due to the limited time and computational resources, we can only show the results of these two tasks. More benchmarking results will be completed and added to the updated manuscript.
>
> [1] ICLR2024-Evaluating representation learning on the protein structure universe

---

> > ### Author Response · Authors · 2024-08-12
> >
> > Thank you again for your hard work in reviewing our paper!
> >
> > We hope you've had a chance to review our responses to your comments. Please let us know if you have any further questions or concerns. We greatly appreciate your feedback and are committed to clarifying the model innovation and the evaluation of the additional benchmarks.

---

> > > ### Comment · Reviewer_qqru · 2024-08-12
> > >
> > > Thanks for your response!
> > >
> > > It is true that the proposed transformer which is based on FA is more efficient than Equiformer which is based on higher-order tensors, which is an advantage. However, to my knowledge, the idea of such architectural design is not so novel. And some important baselines are missing, e.g., [1].
> > >
> > > I wonder the performance of the model on SE(3)-equivariant property prediction tasks. To be more precise, contact map prediction, binding site prediction, fold prediction and reaction prediction are SE(3)-invariant property prediction tasks, while protein-protein docking or protein-ligand docking tasks that I mentioned in the previous review can be formulated as SE(3)-equivariant prediction tasks with well-established datasets and benchmarks. I highly recommend the authors to test the proposed architecture on these tasks.
> > >
> > > My score currently remains the same.
> > >
> > > References:
> > >
> > > [1] Brehmer, Johann, et al. "Geometric algebra transformer." NeurIPS 2023

---

> > > > ### Author Response · Authors · 2024-08-13
> > > >
> > > > ```
> > > > Comments: It is true that the proposed transformer which is based on FA is more efficient than Equiformer which is based on higher-order tensors, which is an advantage. However, to my knowledge, the idea of such architectural design is not so novel.
> > > > ```
> > > > **Ans**: We believe the key contribution and novelty of our architecture lie in demonstrating the potential of designing and applying equivariant modules based on FA, which, to the best of our knowledge, has not been explored previously. Moreover, the primary motivation behind our architectural design is to address the inherent issues of the prior Transformers, as mentioned in Lines 35-38, which is non-trivial and can effectively enhance performance.
> > > >
> > > > ```
> > > > Comments: And some important baselines are missing, e.g., [1].
> > > > ```
> > > > **Ans**: Thank you for bringing this to our attention! We have applied GATr’s official implementation and evaluated its performance on the contact map prediction task, as shown below:
> > > >
> > > > |  |  | EGNN | GATr | FAFormer |
> > > > | --- | --- | --- | --- | --- |
> > > > | Protein-RNA | F1 | 0.1093 | 0.1107 | 0.1284 |
> > > > |  | PRAUC | 0.0964 | 0.0976 | 0.1113 |
> > > > | Protein-DNA | F1 | 0.1208 | 0.1077 | 0.1457 |
> > > > |  | PRAUC | 0.1139 | 0.1075 | 0.1279 |
> > > >
> > > > The input features for GATr are consistent with those used in our method and other baselines, including coordinates and pretrained residue/nucleotide embeddings. Following GATr’s setup, the input coordinates are first mapped to tri-vectors. However, GATr’s original implementation is designed for general point clouds and uses full attention among points, which is memory-inefficient for large biomolecules. To address this, we limited the attention to K-nearest neighbors. The hyperparameters were tuned using the same process outlined in Appendix A and are shown below:
> > > > - Hidden dim: 64
> > > > - Hidden dim for multivector: 32
> > > > - Number of neighbors: 30
> > > > - Number of attention blocks: 5
> > > > - Dropout ratio: 0.2
> > > >
> > > > ```
> > > > Comments: I wonder the performance of the model on SE(3)-equivariant property prediction tasks. To be more precise, contact map prediction, binding site prediction, fold prediction and reaction prediction are SE(3)-invariant property prediction tasks, while protein-protein docking or protein-ligand docking tasks that I mentioned in the previous review can be formulated as SE(3)-equivariant prediction tasks with well-established datasets and benchmarks. I highly recommend the authors to test the proposed architecture on these tasks.
> > > > ```
> > > > **Ans**: Existing docking benchmarks [1,3] primarily compare different docking models, such as diffusion model-based approaches [4] and alignment-based models [2]. FAFormer, as a geometric backbone, is not directly comparable to these docking methods but is **compatible** with them by extending or replacing the geometric encoder they use.
> > > >
> > > > To demonstrate this compatibility, we present the results of integrating FAFormer with Equidock [2] on the protein-protein docking task using our protein complex dataset:
> > > >
> > > > |  | complex RMSD | interface RMSD |
> > > > | --- | --- | --- |
> > > > | Equidock | 20.266 | 22.769 |
> > > > | Equidock w/ FAFormer | 18.572 | 18.089 |
> > > >
> > > > In this table, "complex RMSD" represents the mean distance between each pair of residues, while "interface RMSD" focuses specifically on the binding interface. These preliminary results demonstrate that incorporating FAFormer can effectively enhance the performance of existing docking methods.
> > > >
> > > > We believe that extending FAFormer to different fields, such as structure prediction and docking, is a promising research direction that we plan to explore further. A discussion will be added to our updated manuscript.
> > > >
> > > > [1] Deep Learning for Protein-Ligand Docking: Are We There Yet?
> > > >
> > > > [2] Independent SE(3)-Equivariant Models for End-to-End Rigid Protein Docking
> > > >
> > > > [3] Accurate structure prediction of biomolecular interactions with AlphaFold 3
> > > >
> > > > [4] DiffDock: Diffusion Steps, Twists, and Turns for Molecular Docking

---

> > > > > ### Author Response · Authors · 2024-08-13
> > > > >
> > > > > Thank you once again for your valuable comments on our paper! As the deadline is approaching, we may not be able to respond after that. Please let us know if you have any further questions or concerns. We are committed to addressing your feedback.

---

> > > > > > ### Comment · Reviewer_qqru · 2024-08-14
> > > > > >
> > > > > > Thanks for providing the experimental results! I have two related questions:
> > > > > >
> > > > > > - Why not use the standard protein complex dataset? In the dataset you proposed in the paper, there are only about 4k protein-protein complexes, while in the widely used protein-protein complex dataset, i.e., the Database of Interacting Protein Structures (DIPS), there are about 40k samples. The experiments on such a small dataset may not be very convincing.
> > > > > >
> > > > > > - The results of complex RMSD and interface RMSD are too high. In this case, the numerical results (~18) may indicate invalid docking results. Please refer to the paper of EquiDock for more details. Their results are much lower than the reported in the above response.

---

### Official Review · Reviewer_DGm4 · 2024-07-11

**Soundness:** 3
**Presentation:** 2
**Contribution:** 3
**Rating:** 7
**Confidence:** 3

**Summary:**

The authors propose a novel equivariant model, FAFormer, which leverages the frame averaging operation as an integral geometric component within the Transformer. The authors prove the invariance and equivariance of the architecture. They further conduct experiments showing that FAFormer performs well in contact map prediction and could serve as a strong binding indicator for aptamer screening.

**Strengths:**

- FAFormer is novel in incorporating frame averaging within each layer of the Transformer. The FAlinear module provides a novel method for creating invariant models.

- Empirical results demonstrate the superior performance of FAFormer on contact map prediction and aptamer screening tasks. FAFormer excels in screening aptamers and outperforms AlphaFold3.

- The authors provide a detailed proof of the invariance and equivariance property.

**Weaknesses:**

- The F1 score for contact map prediction tasks is about 0.1, which is very low. I question whether this score can be used to judge which model is better. The model potentially performs worse due to separate modeling of nucleic acid and protein, the linear prediction head, and the difficulty of the pairwise prediction task. I suggest involving other node-level and graph-level prediction tasks for comparing FAFormer with other equivariant neural networks.

- The main idea of the model is confusing. If we already know the coordinates of the protein and nucleic acid, why do we need further encoding and prediction for contact prediction, instead of directly counting the contacts?

**Questions:**

- The module structures with formulas are a bit difficult to follow. Also, could you provide more intuition on the design of these modules?

- Since we are constructing a KNN graph and always leveraging the coordinate X and edge E information, why should the model be called a “former” instead of a GNN, which could be misleading?

- As far as I know, RosettaFoldNA is used mainly for protein and NA structure prediction. However, FAFormer uses coordinate information as input. Is that a fair comparison?

- I am curious about the efficiency of FAFormer compared with other equivariant NNs. Could you provide some analyses?

- During experiments, do other methods include the same featurization? And are you training all models from scratch?

**Limitations:**

The work provides an innovative framework, FAFormer, and performs well on contact map prediction and aptamer screening tasks. However, the experimental setting is not convincing enough and requires refinement.

---

> ### Author Rebuttal · Authors · 2024-08-06
>
> ```
> Weakness: The F1 score for contact map prediction tasks is about 0.1, which is very low. I question whether this score can be used to judge which model is better. I suggest involving other node-level and graph-level prediction tasks for comparing FAFormer with other equivariant neural networks.
> ```
> **Ans**: We would like to clarify that no matter the contact map prediction performance, the **main focus** of this paper is unsupervised aptamer screening. Contact map prediction is proposed as a prerequisite task due to its strong correlation with the screening task. While contact map prediction is challenging due to sparse labels, all comparisons are fair, with the same input features and learning pipelines. For benchmarking:
>
> - We have provided results on **binding site prediction** in *Appendix E.1*. This task predicts the NA-binding residues for a given protein, which is a **node-level task**. The metrics exceed 0.4 since this task is much easier than the contact map prediction.
> - The **aptamer screening task** is a **graph-level task** aimed at retrieving the positive RNAs for a protein.
>
> Moreover, we present FAFormer’s performance on two protein-related tasks:
>
> - Fold prediction: Predict the fold class (1195 classes) for protein, with three test sets (Fold, Family, and Superfamily). Accuracy is the evaluation metric.
> - Reaction prediction: Predict the reaction class catalyzed by a given protein (384 classes). Accuracy is the evaluation metric.
>
> Following the experimental settings of ProteinWorkshop[1], including dataset splits and Ca-only input features, the comparison results are shown below. FAFormer demonstrates the best performance across most tasks:
>
> ||Fold(Fold)|Fold(Family)|Fold(Superfamily)|Reaction|Average|
> |---|---|---|---|---|---|
> |SchNet|0.2071|0.7670|0.2375|0.5894|0.4502|
> |GearNet|0.3090|0.9340|0.4465|0.7814|0.6177|
> |EGNN|0.2577|0.9193|0.3568|0.6578|0.5479|
> |GCPNet|**0.3276**|0.9359|0.4110|0.6697|0.5860|
> |TFN|0.2512|0.9188|0.3426|0.6922|0.5512|
> |FAFormer|0.2451|**0.9661**|**0.4883**|**0.7970**|**0.6241**|
>
> Due to limited time and computational resources, we only show these two tasks' results. More results will be completed and added to the updated manuscript.
>
> [1] ICLR2024-Evaluating representation learning on the protein structure universe
> ```
> Weakness: If we already know the coordinates of the protein and nucleic acid, why do we need further encoding and prediction for contact prediction, instead of directly counting the contacts?
> ```
> **Ans**: We use the individually **predicted structures** of proteins/nucleic acids during the evaluation (*Lines 207-209*), and all the input structures are **decentralized** (*Line 597*), meaning the coordinates are moved to the zero centers to avoid label leakage. Consequently, the distances between two input structures cannot be used to directly indicate the complex's contacts.
>
> ```
> Question: Could you provide more intuition on the design of these modules?
> ```
> **Ans:** FA is a general framework that endows a given encoder with equivariance, allowing for the flexible design of equivariant modules. Using FA as a module rather than a model wrapper also avoids an 8x increase in computation. We are glad to elaborate more on each module’s intuition:
>
> - Local Frame Edge Module: In biomolecules, atoms interact through chemical bonds and electronic forces. Embedding these pairwise relationships allows the model to **represent these intricate dependencies** explicitly. We **differentiate the spatial context** around each node by considering the local point cloud centered on each target node. FA embeds the directed vectors between source and target nodes, capturing the direction of interactions.
> - Biased MLP Attention Module: The attention mechanism is extended to include edge representations, biasing the attention map to **prioritize/deprioritize certain weights** based on interactions. Node coordinates are also updated to **simulate conformational changes** during molecular interactions, which is crucial for optimal binding.
> - Global Frame FFN: We extend the FFN by incorporating FA to update representations in a geometry-aware context. The attention between two atom representations, combined with coordinate information, **functions similarly to distance calculations**.
>
> We will add these discussions to the revised manuscript.
> ```
> Question: Why should the model be called a “former” instead of a GNN, which could be misleading?
> ```
> **Ans**: The core module of FAFormer is the biased attention module, which is the primary reason for considering our model a Transformer. We limit the attention to k-nearest neighbors to reduce the computation, which is commonly used in previous geometric Transformers.
> In the paper (*Lines 90-94*), we have summarized that FAFormer is a hybrid of GNN and Transformer, as the edge module functions similarly to GNN aggregation.
> ```
> Question: As far as I know, RosettaFoldNA is used mainly for protein and NA structure prediction. However, FAFormer uses coordinate information as input. Is that a fair comparison?
> ```
> **Ans**: Besides the input sequence, RosettaFoldNA utilizes **searched MSAs** and **protein structure templates** as input. The protein structure templates are the coordinates of the MSAs. In contrast, the input structures for FAFormer are predicted solely based on input sequences.
> ```
> Question: I am curious about the efficiency of FAFormer compared with other equivariant NNs.
> ```
> **Ans**: We have provided an analysis in *Appendix B*, which includes computational complexity and wall-clock time comparisons. In summary, our proposed FAFormer demonstrates greater efficiency than spherical harmonics and FA-based Transformers.
> ```
> Question: Do other methods include the same featurization? And are you training all models from scratch?
> ```
> **Ans**: Yes, all the baseline models use the same features and are all trained from scratch. The hyperparameter and training details are provided in *Appendix A*.

---

> > ### Author Response · Authors · 2024-08-12
> >
> > Thank you again for your hard work in reviewing our paper!
> >
> > We hope you've had a chance to review our responses to your comments. Please let us know if you have any further questions or concerns. We greatly appreciate your feedback and are committed to addressing any potential issues.

---

> > > ### Comment · Reviewer_DGm4 · 2024-08-12
> > >
> > > Thank you for the responses. I believe all of my concerns have been addressed, and I look forward to seeing additional results/discussion in the updated manuscript.
> > >
> > > I have raised my score from 5 to 7.

---

> > > > ### Author Response · Authors · 2024-08-13
> > > >
> > > > We greatly appreciate your valuable comments on our paper! All the suggestions will be added to our updated manuscript.

---

### Author Rebuttal · Authors · 2024-08-06

We appreciate the reviewers for noting that we propose a novel model (DGm4,npWJ) to address a meaningful problem (DGm4,qqru,npWJ) with a comprehensive evaluation (DGm4,npWJ). We further summarize our key contributions as follows:

1. We explore a new angle to conduct aptamer screening in an unsupervised manner by leveraging the strong correlation with the contact map prediction task.
2. We propose a new equivariant Transformer architecture, FAFormer, by integrating Frame Averaging (FA) within each module. FA as an integral component offers the flexibility to design expressive and equivariant modules, highlighting a new possibility for geometric encoder design in this domain.
3. We construct three protein complex datasets (Protein-RNA/DNA/Protein) and five aptamer datasets to evaluate the models. Our proposed architecture achieves SOTA performance over baselines, including RoseTTAFoldNA and AlphaFold3.

Additionally, two protein understanding tasks are included to further demonstrate the effectiveness of FAFormer (DGm4,qqru). We thank all the reviewers for their valuable comments, and the corresponding refinements will be added to our updated manuscript.

---

### Decision · Program_Chairs · 2024-09-25

**Decision:**

Accept (poster)

**Comment:**

This work introduces FAFormer, a novel equivariant transformer architecture that seamlessly integrates frame averaging within each transformer block. By incorporating FA as a core component, the architecture allows for the design of expressive and equivariant modules. Results demonstrate that FAFormer outperforms existing equivariant models in contact map prediction across three protein complex datasets. The reviewers found this framework generally interesting and innovative, with valuable yet underexplored application scenarios. Despite some minor concerns, this is a strong paper that I recommend for acceptance to NeurIPS.